# Pectin Peek-a-Boo: Homogalacturonan Turnover During Flooding-Induced Legume Root Aerenchyma Formation

**DOI:** 10.3390/plants14172620

**Published:** 2025-08-23

**Authors:** Timothy J. Pegg, Daniel K. Gladish, Robert L. Baker

**Affiliations:** 1Department of Biology, Midwestern State University, Wichita Falls, TX 76308, USA; 2Department of Biological Sciences, Miami University, Hamilton, OH 45011, USA; gladisdk@miamioh.edu; 3Life Sciences Program, University of Wyoming, Laramie, WY 82071, USA; rob.baker@uwyo.edu

**Keywords:** aerenchyma, cell wall, hypoxia, pectin, programmed cell death, immunolabeling, de-methyl-esterification

## Abstract

Flooding can cause root hypoxia and can lead to significant agricultural losses. Therefore, understanding plant adaptations to flooding, including root aerenchyma development, is one important avenue for insuring future global food security. We investigated cell wall modifications during root aerenchyma formation in response to the prolonged 0–48 h flooding of *Phaseolus coccineus*, *Pisum sativum*, and *Cicer arietinum* seedlings. Using transmission electron microscopy, toluidine blue O (TBO) staining, and immunolabeling with antibodies targeting de-methyl-esterified homogalacturonan (DMEH), partially DMEH, and methyl-esterified homogalacturonan (MEH), we examined changes in cell wall composition. Transmission electron microscopy and TBO staining revealed degradation of cell walls and middle lamella, with accumulation of DMEH near flooding-induced aerenchyma cavities. Immunolabeling indicated increased DMEH epitope availability in flooded roots, suggesting a role in cell wall remodeling. Enzyme pretreatments, used to “unmask” homogalacturonan by removing cellulose and hemicellulose, revealed that specific forms of homogalacturonan, particularly DMEH complexed with calcium and MEH, are masked by these cell wall components. The study highlights the complex interplay of pectin, cellulose, and hemicellulose in cell wall degradation during aerenchyma development, providing insights into legume flooding stress responses.

## 1. Introduction

By 2050, the global demand for food is predicted to grow by 70–85% [1]. Climate change will threaten the security of food supply by causing temperature extremes, erratic rainfall patterns, and shifts in our biosphere’s suitability for modern agriculture [2,3]. Low oxygen conditions (i.e., hypoxia) from prolonged flooding have the potential to cause significant agricultural losses [4,5]. Therefore, understanding plant adaptations to flooding is one important avenue for ensuring future global food security.

Flooding affects several aspects of plant root physiology and morphology. Hypoxic conditions due to flooding may inhibit aerobic respiration and subsequent ATP production in root tissues [6]. Areas of high metabolic activity, such as root tips, are particularly vulnerable to sustained hypoxic conditions and will experience decreased growth or possible death of primary root tissue [7]. This situation may be exacerbated by changes in soil pH and redox potential that negatively influence the uptake of phosphorus, nitrogen and other nutrients into plant roots [7]. Plants may respond to flooding stress by altering ethylene signaling pathways to induce adventitious root growth near the water surface, thereby accessing oxygen and potentially replacing damaged primary roots [6,7]. Anaerobic respiration pathways, such as ethanol fermentation, may also be utilized in root tissues to temporarily compensate for oxygen restrictions until flooding subsides or toxic byproduct accumulation results in cellular damage [8].

Aerenchyma development is another adaptation to flooding [9]. Aerenchyma in roots allows plants to tolerate flooding-induced hypoxia by serving as air-filled cavities that maintain sufficient oxygen levels for cellular respiration. Aerenchyma also reduces the number of cells utilizing oxygen within the root cortex or stele tissues [9,10,11,12]. Aerenchymas are categorized as either schizogenous aerenchyma, which are formed by differential growth that does not involve programmed cell death (PCD) [13,14] or lysigenous aerenchyma. Lysigenous aerenchyma are formed by remodeling cell wall components during PCD. Previous research in cool-season legumes [15,16], *Oryza sativa* (rice) [17], and *Zea mays* (maize) [18] indicates that flooding-induced lysigenous aerenchyma formation may rely on the modification of pectins within the primary cell wall.

The primary cell wall consists of a complex layer of polysaccharides and proteins on the outer surface of the plasma membrane that encases the entire plant cell. Primary cell walls of dicotyledonous and non-graminaceous monocotyledonous plants are characterized by an interlinked matrix of cellulose microfibrils and xyloglucan in a hydrated pectin network [19]. The primary backbone structure of pectin is homogalacturonan, a repeating monomer of α-1,4-linked D-galacturonic acid that is initially produced from the cellular Golgi complex as highly MEH residues [20]. Pectin methylesterases (PMEs) may act randomly on the methyl ester groups of homogalacturonans, resulting in DME, which releases protons that then promote the action of endopolygalacturonases and contribute to cell wall loosening in plant primary cell walls [21]. Methyl ester groups may also protect homogalacturonan from degradation by shielding linkages between D-galacturonic acid residues from hydrolytic enzyme cleavage [21]. DME modifies homogalacturonan by removing these protective methyl ester groups and permitting enzyme-mediated degradation of cell walls [22]. DME is a requirement for many plant developmental processes, such as leaf abscission, floral organ abscission, fruit ripening, and lysigenous aerenchyma formation [23,24,25,26].

However, the extent of homogalacturonan modification in roots with vascular aerenchyma formation remains unclear. Previous work suggests that homogalacturonan with varying degrees of DME is localized in cell layers immediately adjacent to the developing aerenchyma [16]. DMEH may also occur throughout the root stele, where it prepares cell walls to be degraded when they come into contact with the expanding borders of developing aerenchyma. DMEH distal from expanding aerenchyma may be “masked” from the antibodies used to detect it by other cell wall components, thereby explaining why DMEH has not been identified in previous immunolabeling experiments [15,16]. It is well established that hemicelluloses (e.g., xyloglucan) form complexes with cellulose [27] and homogalacturonan may bind to cellulose to stabilize cell wall structures [28,29]. It is possible that these cell wall interactions prevent DMEH not immediately adjacent to developing aerenchyma cavities from being antibody-labeled.

Our objective was to identify the true extent of homogalacturonan DME in roots during flooding-induced aerenchyma formation. Our hypothesis was that removing cellulose and hemicellulose by enzyme pretreatment would “unmask” homogalacturonan with varying degrees of DME throughout root tissues and would reveal the extent of these epitopes present in cell walls adjacent and distal to incipient aerenchyma.

## 2. Results

### 2.1. Ultrastructure Analysis and Histochemical Labeling of Cell Walls Indicates Changes in Homogalacturonan Composition

TEM analysis showed that, within 24 h of flooding, the cell wall fragments and middle lamella of *Phaseolus coccineus*, *Pisum sativum*, and *Cicer arietinum* samples were reduced or eliminated, and darkly stained degraded cell wall components accumulated (Figure 1A–C). Similar cell wall and middle lamella degradation was not present in non-flooded treatments (Figure 1G–I). These data suggested that degradation may release DMEH to be bound by osmium tetroxide-ruthenium red stain.

Potential changes in cell wall chemistry between non-flooded and flooded root samples were evaluated by observing alterations in toluidine blue staining (Figure 2). Both non-flooded and 24 h-flooded samples showed metachromatic staining typical of toluidine blue with the lignified xylem cell walls (green/green-blue), phloem sector fibers (bright blue), cortical parenchyma, and central parenchymatous tissue (pink) easily distinguishable as different colors (Figure 2). However, 24 h-flooded samples displayed bright magenta staining near the aerenchyma cavities that was not observed in non-flooded samples (Figure 2A,C,E). The bright magenta coloration near the aerenchyma of plants exposed to flooding conditions (Figure 2B,D,F) may indicate the presence of negatively charged functional groups on cell wall components, such as DMEH residues from degraded cell walls [30]. Supporting evidence for DMEH degradation was acquired when accumulated D-galacturonic acid residues, likely cleaved from the homogalacturonan backbone structure, were observed in legume roots from 12–48 h flooding timepoints, whereas these residues were absent at 0 h non-flooding timepoints (Appendix A). The toluidine blue and D-galacturonic acid assays aligned with the ruthenium-red staining TEM micrographs (Appendix A), indicating the possible presence of negatively charged pectin domains, such DMEH, in degrading cells bordering aerenchyma cavities.

### 2.2. Immunolabeling Suggests DMEH Occurs in Cell Walls Bordering Aerenchyma

We used the 2F4 antibody (Figure 3), which specifically binds to DMEH-calcium complexes [31]. Non-flooded *P. coccineus* and *P. sativum* (Figure 3E,I) had strong, uniform antibody labeling of the cortical parenchyma, endodermis, pericycle, and central parenchymatous tissue. By comparison, non-flooded samples of *C. arietinum* (Figure 3A) displayed very weak and inconsistent antibody labeling within the cortical parenchyma and central parenchymatous tissue. Differences in the location of DMEH in non-flooded samples implied that the *C. arietinum* cell wall composition qualitatively differs from that of *P. coccineus* and *P. sativum*. All flooding time points for *P. coccineus*, *P. sativum*, and *C. arietinum* (Figure 3B–D,F–H,J–L) possessed similar 2F4 binding patterns, which indicated that DMEH occurs in the cortex, pericycle, endodermis and central parenchyma of these species. The results suggested that substantial homogalacturonan turnover occurs during aerenchyma formation, resulting in the exposure of more DMEH-calcium domains to antibody binding.

### 2.3. Unmasking Cell Wall Homogalacturonan Reveals the Extent of Pectin Modification in Roots Exposed to Flooding

To evaluate the extent and degree of homogalacturonan DME in legume roots during aerenchyma formation, root cross-sections were pretreated with enzymes to remove other cell wall components that might block antibody labeling of homogalacturonan. We refer to enzymatic removal of cell wall components to reveal antibody binding sites as “unmasking.” Unmasked root samples were labeled with 2F4 (antibody binding to DMEH complexed with calcium ions), JIM5 (antibody binding to partially MEH), and JIM7 (antibody binding to MEH) pectin antibodies to identify the patterns of homogalacturonan domain distribution within legume roots subjected to 24 h-long periods of flooding.

Samples labeled with 2F4 were used to determine if DMEH complexed with calcium ions could be found in cell walls both proximal and distal from developing root aerenchyma. Compared to sodium citrate treatment (Figure 4A,F,K), which served as a control for 2F4 antibody labeling without enzymatic unmasking, enzyme pretreatments that removed hemicelluloses (xylanase) resulted in minimal changes to the binding pattern of the 2F4 antibody (Figure 4D,I,N). By contrast, pretreatment with cellulase (which removes cellulose) resulted in increased antibody binding within the central parenchyma of *P. coccineus* and *P. sativum* samples (Figure 4C,H) and increased antibody binding in cells immediately adjacent to the aerenchyma in *C. arietinum* (Figure 4M) compared to surrounding tissues and the sodium citrate control (Figure 4A,F,K). Interestingly, cellulase pretreatments resulted in the loss of 2F4 antibody labeling of xylem cell walls within *P. coccineus* (Figure 4C). The pectinase pretreatment resulted in antibody binding within the endodermis of *P. coccineus* (Figure 4E) compared to the sodium citrate control. For *P. sativum* and *C. arietinum* pectinase pretreatments, no difference in binding patterns (Figure 4J,O) was observed compared to the sodium citrate pretreatment (Figure 4A,K). The results suggested that binding of the 2F4 antibody to DMEH complexed with calcium ions was generally unaffected by the pectinase pretreatment assay.

Samples labeled with JIM5 were used to determine if unmasking influenced the antibody binding patterns to partially DMEH in root tissues (Figure 5). Compared to the sodium citrate buffer negative control pretreatment for lack of enzymatic unmasking (Figure 5A,F,K), the JIM5 antibody binding patterns were maintained after sodium carbonate treatment for artificially-induced homogalacturonan DME across the surface of the sample sections and after cellulase and xylanase experimental treatments for each species (Figure 5B–N). Both *P. coccineus* and *P. sativum* displayed strong antibody labeling in the central parenchyma and cortex. In both species, there was notable labeling of partially DMEH in cells adjacent to the developing aerenchyma (Figure 5A–D,F–I). *C. arietinum* also had JIM5 antibody labeling in the central parenchyma and cortex, but the fluorescence intensity did not increase in binding near the developing aerenchyma. Pectinase pretreatments resulted in the loss of signal for all three species compared to the sodium citrate control, with binding diminished or absent from the central parenchyma and cortex (Figure 5E,J,O). However, JIM5 antibody labeling remained in the outer boundaries of xylem cell walls and phloem within *P. coccineus* and *P. sativum* (Figure 5E,J) and endodermis and inner cortex of *C. arietinum* (Figure 5O) compared to the sodium citrate control. These results indicated that *P. coccineus* and *P. sativum* have similar distributions of partially DMEH near aerenchyma, and these distributions differ from that of *C. arietinum*. In addition, the consistency of JIM5 antibody binding across most enzyme pretreatments implied that partially DMEH was not masked by the cell wall components tested in this study.

Cell walls containing MEH were labeled with the JIM7 antibody to evaluate differences in patterns of MEH, compared to 2F4 and JIM5-labeled samples, that may indicate unmasking of multiple homogalacturonan epitopes in flooded aerenchyma-forming legume roots. JIM7-labeled samples (Figure 6) that were pretreated with sodium citrate (Figure 6A,F,K) displayed antibody binding of the xylem cell walls, endodermis, and inner cortex that was generally unchanged across all enzyme pretreatments for *P. sativum* and *C. arietinum* (Figure 6G–J,L–O). By contrast, after cellulase and xylanase treatments, *P. coccineus* displayed loss of JIM7 binding to the central parenchyma but stronger binding to cells adjacent to aerenchyma cavities (Figure 6C,D). In addition, sodium carbonate pretreatment of *P. coccineus* led to stronger binding of the JIM7 antibody to xylem cell walls and phloem (Figure 6B) compared to the sodium citrate control treatment (Figure 6A). For *P. coccineus*, the pectinase pretreatment led to very weak and inconsistent antibody labeling throughout the root section (Figure 6E). This indicated that MEH is masked by cellulose and xylan in *P. coccineus* (Figure 6A–E) but is not masked in the roots of *P. sativum* and *C. arietinum* (Figure 6F–O).

The overall results of the pretreatment assay suggested that DMEH-calcium complexes (2F4), partially DMEH (JIM5), and MEH (JIM7) antibody binding are generally unaffected by enzyme pretreatments. However, cellulase pretreatments increased 2F4 antibody binding in *P. coccineus* and *C. arietinum*, and both cellulase and xylanase pretreatments increased JIM7 labeling in *P. coccineus*, which indicated that limited cellulose and xylanase unmasking of homogalacturonan occurred in at least two legume species.

## 3. Discussion

Agricultural losses due to increased prolonged flooding are a potential consequence of global climate change. Elucidating the mechanistic basis of plant flooding adaptations, such as root aerenchyma formation, offers a potential avenue for crop improvement to counteract these losses. Lysigenous root aerenchyma development involves the remodeling and degradation of root cell walls through programmed cell death (PCD) to create air-filled cavities that protect tissues from hypoxia caused by water immersion [13,32]. To successfully form aerenchyma, cell wall remodeling requires modification of the pectin backbone structure, homogalacturonan, through DME [18]. DME results in degradation of homogalacturonan by permitting cleavage of hydrolytic bonds between α-1,4-linked D-galacturonic acid residues by pectic enzymes such polygalacturonase and pectin lyase [33,34,35]. While it is known that DMEH is found in cell walls adjacent to developing aerenchyma [16], it is unclear if it may be found in other tissues of roots subjected to flooding. We analyzed whether DMEH is masked by other cell wall components in tissues not actively participating in aerenchyma formation and whether unmasking is a component of cell wall remodeling during to cavity formation. We found evidence that upon removal, or “unmasking,” of cellulose and hemicellulose (xylan) with enzyme pretreatments, homogalacturonan with specific degrees of DME was found throughout the roots of *P. sativum*, *P. coccineus*, and *C. arietinum* that were subjected to flooding. Combined with enzyme activity assays and previous immunolabeling data [16], our findings suggest that unmasking of cell wall homogalacturonan in tissue layers near aerenchyma may occur during cavity formation but does not directly influence the degree of DMEH.

In this study, chemical modifications to the structure of cell wall homogalacturonan were identified in cells bordering aerenchyma cavities across three legume species. Osmium tetroxide-ruthenium red labeling patterns in TEM sections displayed increased availability of DMEH in 24 h flooding treatments compared to non-flooded samples (Figure 1). The presence of DMEH provided evidence of cell wall remodeling and degradation typical of lysigenous aerenchyma development [36]. These data were further supported by immunolabeling data for DMEH complexed with calcium ions in a flooding time-course (Figure 3). Labeling with toluidine blue, a differential metachromatic stain, displayed bright magenta coloration near aerenchyma cavities of flooded samples (Figure 2B,D,F) compared to a paler magenta coloration of central parenchyma observed in non-flooded samples (Figure 2A,C,E). The magenta stain observed in walls adjacent to developing aerenchyma cavities may indicate increased negative functional group availability, which often results from liberation of DME pectin residues during degradation of cell walls and middle lamella [37,38]. Liberation of DMEH residues was confirmed through measurement of α-1,4-linked D-galacturonic acid residues in root samples across a 0–48 h flooding timepoint series (Appendix A). These results suggest that DMEH needs to be degraded for successful aerenchyma formation.

To evaluate whether other cell wall matrix components may block or protect homogalacturonan from degradation during aerenchyma formation, we performed enzyme pretreatment assays to remove specific cell wall components surrounding homogalacturonan. The removal of cell wall components permitted antibody labeling, or enzyme action, upon a specific chemical substrate, which is referred to as “unmasking.” Unmasking events have been observed to permit antibody binding of cell wall xyloglucan in *Arabidopsis thaliana* [39] and pectin methylesterase activity upon homogalacturonan methyl-ester groups of *Lycopersicon esculentum* [40]. In this study, enzyme pretreatments served to reveal the extent of DMEH, partially DMEH, and MEH epitopes present in cell walls adjacent and distant from aerenchyma cavities. Our data suggest that cellulose and xylan may be removed or modified during aerenchyma formation to unmask two different forms of homogalacturonan: DMEH complexed with calcium ions and completely methyl-esterified pectin. The absence of enzymatic unmasking in JIM5-labeled samples (Figure 5) implies that cellulose/xylose may not associate with partially MEH during aerenchyma formation, thereby permitting antibody labeling to occur throughout root samples regardless of enzyme labeling. These results suggest that only homogalacturonan with specific degrees of DME may be masked by other cell wall polymers. In addition, the repetitious nature of antibody binding patterns across enzyme treatments implies that the modification of homogalacturonan may be independent of cellulose and hemicellulose removal during aerenchyma formation. However, the spatiotemporal organization of cell wall alterations may be species dependent, as indicated by antibody labeling differences between *P. coccineus*, *P. sativum*, and *C. arietinum*.

Our work raises questions regarding the temporal nature of unmasking in legume root aerenchyma. It is unclear if removal of cell wall polymers in tissues adjacent to aerenchyma cavity formation occurs concurrently with modification of homogalacturonan or precedes changes in pectin chemistry. For example, cellulose masking of 2F4 binding sites near aerenchyma borders implicates cellulose removal as a possible component of aerenchyma formation separate to the DME process in all three legume root samples (Figure 4). It is also unknown if the removal of cellulase and xylan components occurs simultaneously, or one precedes the other, during unmasking reactions. Future work combining live-imaging of root tissues and labeling of cell wall enzymes with photoactivatable probes [41,42] would enable real-time visualization of enzymatic degradation and the ability to establish a chronology of developmental events in aerenchyma.

The regulation of gene expression leading to the unmasking of homogalacturonan is also unclear. It is known that hydrogen peroxide (H_2_O_2_) and ethylene signaling pathways play a role in the formation of lysigenous aerenchyma in *P. sativum* and *Arabidopsis thaliana* [43,44,45]. These pathways are initiated by hypoxia from flooding conditions and result in gene expression for cell wall remodeling enzymes, such as hemicellulases (xylanase, xyloglucanase, etc.) and cellulase [46]. Future experiments may find correlations among elevated cellulase and hemicellulase (i.e., xylanase and xyloglucanase) levels and gene expression during aerenchyma development, thereby providing a mechanism to regulate aerenchyma formation.

In conclusion, our study investigated potential unmasking of cell wall pectin during aerenchyma formation in cool-season legume species such as *Phaseolus coccineus*, *Pisum sativum*, and *Cicer arietinum*. We found evidence that selective removal of cellulose and hemicellulose through artificial enzyme application may have implications on pectin detection, and this procedure may mimic the processes that occur during cavity formation.

Combined with previous research [16,47], our data indicate that cool-season legume aerenchyma formation may rely on a complicated interaction of selected pectin, cellulose, and hemicellulose epitopes to enable cell wall degradation required for aerenchyma cavity expansion. Future investigation into the cell wall remodeling events of aerenchyma formation could lead to understanding the regulatory events of legume flooding stress responses and contribute to developing crop species with a higher tolerance to flood-induced hypoxia.

## 4. Materials and Methods

### 4.1. Seedling Growth and Flooding Treatment

Seedlings were grown according to Gladish and Niki [31]. For each species, 40 seeds (*Pisum sativum* and *Cicer arietinum*) or 20 seeds (*Phaseolus coccineus*) were sown per container in 2 L beakers filled with 1800 mL of sterile, super-coarse vermiculite (Perlite Vermiculite Packaging Industries, Inc., North Bloomfield, OH, USA). The vermiculite was moistened with deionized water and covered. Beakers were placed in 25 °C growth chambers for 5 d in complete darkness to initiate root growth. Three replicates for each flooding treatment (0, 12, 24, and 48 h water immersion) were created using a separate 2 L beaker for each replicate. To perform flooding treatments, four sets of three 2 L beakers (0, 12, 24, and 48 h water immersion) were removed from the growth chambers, placed under a laminar flow hood, and filled with sterile deionized water to the surface level of the vermiculite substrate. One set of non-flooded beakers representing the 0 h time point was harvested at that time. The remaining beakers were returned to the 25 °C growth chambers and removed at either 12 h, 24 h, or 48 h after flooding to be harvested for sectioning.

### 4.2. Sectioning, Fixation, and Embedding

Sectioning, fixation, and embedding of samples was performed according to methods by Pegg et al. [16]. Briefly, 10 root segments were harvested from each species per flooding timepoint or non-flooding treatment. Segments were cut with razor blades (Electron Microscopy Services, Hatfield, PA, USA) from either 1.5–5 cm (*Pisum sativum* and *Phaseolus coccineus*) or 3–7 cm (*Cicer arietinum*) away from root tips. Segments were fixed in 1% paraformaldehyde and 2% glutaraldehyde solution in deionized water in 0.1 M sodium cacodylate buffer, pH 7.4, at 5 °C. Segments were then washed three times with deionized water (15 min per wash), embedded in 3.5% agarose (Sigma-Aldrich, CAS 9012-36-6, St. Louis, MO, USA) at 40 °C, solidified, and mounted on epoxy resin stubs. Samples were sectioned at 150 µm thickness using a Vibratome Series 1000 Sectioning System (Ted Pella, Inc., Redding, CA, USA). Sections from each root were stored separately in three separate pools (per treatment, per species) in 0.1 M tris-buffered saline solution (pH 7.4) with 0.1% sodium azide at 5 °C.

### 4.3. Immunolocalization

Ten sections from each species pool, for non-flooded (5 sections) and flooded treatments (5 sections), were placed into sterile 24-well cell culture plates and blocked with 7% normal goat serum (Thermo Fisher Scientific, Waltham, MA, USA) for 24 h at 5 °C. The samples were washed 3× (15 min per wash) with 10 mM Tris-buffered saline (pH 7.4) containing 0.1% TWEEN-20 (TBST) and then incubated with 1/20 dilutions of 2F4, JIM7, or JIM5 (CCRC, University of Georgia, USA) monoclonal antibodies for 24 h at 5 °C (Table 1). After incubation, the samples were washed three times with TBST buffer and treated with 1/500 dilutions of IgG goat anti-rat secondary antibody conjugated to Alexa Fluor™ 647 fluorescent dye (Thermo Fisher Scientific, USA) for 24 h at 5 °C while wrapped with Parafilm M sealing film and covered in aluminum foil. Samples were washed a final time with three changes of TBST buffer and mounted in 100% glycerol (Sigma-Aldrich, CAS 56-81-5, USA) on standard 1 mm glass slides. Slides were covered with 24 × 60 mm No. 1 coverslips with two 22 × 22 mm No. 1 coverslips applied underneath to serve as spacers. Samples were stored at 5 °C in darkness when not in use.

### 4.4. Enzyme Pretreatment Assays

The enzyme pretreatment assays were modified from previous methods [16,48,49]. Unsorted root sections selected from the 24 h flooding timepoint of each legume species were separately incubated in the following enzyme solutions (Millipore Sigma, Burlington, MA, USA) at 50 °C for 2 h: 4% cellulase, 1% xylanase, 2% xyloglucanase, and 3% pectinase in 0.05 M citrate buffer (pH 5.0). Treatment with 0.1 M sodium carbonate (pH 11.4) at 50 °C for 2 h was utilized to fully de-methyl-esterify homogalacturonan on exposed surfaces of the sample and ensure binding by the primary antibodies (Table 1). Incubation of samples in 0.05 M citrate buffer (pH 5.0) at 50 °C for 2 h replicated the primary antibody binding patterns for 2F4, JIM5, and JIM7, as observed without enzyme pretreatments. Samples were washed three times with TBST buffer, treated with a monoclonal antibody, and incubated with secondary antibodies conjugated to Alexa Fluor^®^ 647 (Thermo Fisher Scientific, Waltham, MA USA) prior to mounting in 100% glycerol on 1 mm glass slides covered with 24 × 60 mm No. 1 coverslips.

### 4.5. Fluorescence Microscopy

Samples from immunolocalization and the enzyme pretreatment assays were observed using the Olympus FV500 Laser Scanning Confocal system (Olympus Corporation, Bartlett, TN, USA) with 20×/0.70 NA and 40×/0.75 NA dry objectives. Excitation of aldehyde-enhanced autofluorescence and Alexa Fluor^®^ 647 dye was achieved with 405 nm and 633 nm laser diodes, respectively. Images were recorded using a Photometric HQ cooled CCD camera (Teledyne Photometrics, Tucson, AZ, USA).

### 4.6. Cell Wall Degradation Activity Assay

Approximately 25 individual roots were harvested per flooding timepoint (12 h, 24 h, and 48 h) for each legume tested in this experiment (*P. sativum*, *P. coccineus*, and *C. arietinum*). An additional 25 root tissue samples were harvested from non-flooded seedlings (0 h timepoint) from each species to serve as a negative control for aerenchyma formation. Sample collection was repeated three times, for a total of 3 g of tissue per timepoint. Samples were stored at −30 C until use. To begin the activity assay, tissue was frozen in liquid nitrogen and crushed in pre-chilled mortar prior to extraction according to the protocol for the *D-glucuronic*/*D-Galacturonic Acid Assay Kit* (Megazyme Ltd., Bray, Ireland). Samples were placed into 96-well plates and absorbance data for each timepoint were collected via a SpectraMax iD5 microplate reader (Molecular Devices, San Jose, CA, USA) detecting at 340 nm. Data were plotted against a standard curve and used to construct statistical graphs with Microsoft Excel software to evaluate cell wall degradation in terms of pectin deconstruction.

### 4.7. Transmission Electron Microscopy

Root tissue samples from non-flooded and 24 h-flooded timepoints were harvested from *Cicer arietinum*, *Phaseolus coccineus*, and *Pisum sativum*. Samples were sectioned into 1 mm segments and placed in 1% paraformaldehyde and 2% glutaraldehyde fixative in 0.1 M sodium cacodylate buffer (pH 7.4) for 24 h. Samples were washed in distilled water three times, and then stained with 2% osmium tetroxide in 0.1 M sodium cacodylate buffer (pH 7.4) for 24 h. Samples were washed another three times with distilled water, exposed to a graded ethanol dehydration series, and then placed into 1:3, 1:1, and 3:1 ratios of 100% ethanol to Spurr’s resin. Samples were placed in Spurr’s resin to infiltrate for 48 h, and then sectioned to 20 µm thickness with an ultramicrotome. Samples were gathered onto 200 mesh copper grids and stained with Reynold’s lead citrate [50], 0.5% uranyl acetate, and 1% ruthenium red solutions. Micrographs were recorded at 20,000× and 50,000× magnifications using a JOEL JEM-1200EX II TEM at 120 KeV.

## Figures and Tables

**Figure 1 plants-14-02620-f001:**
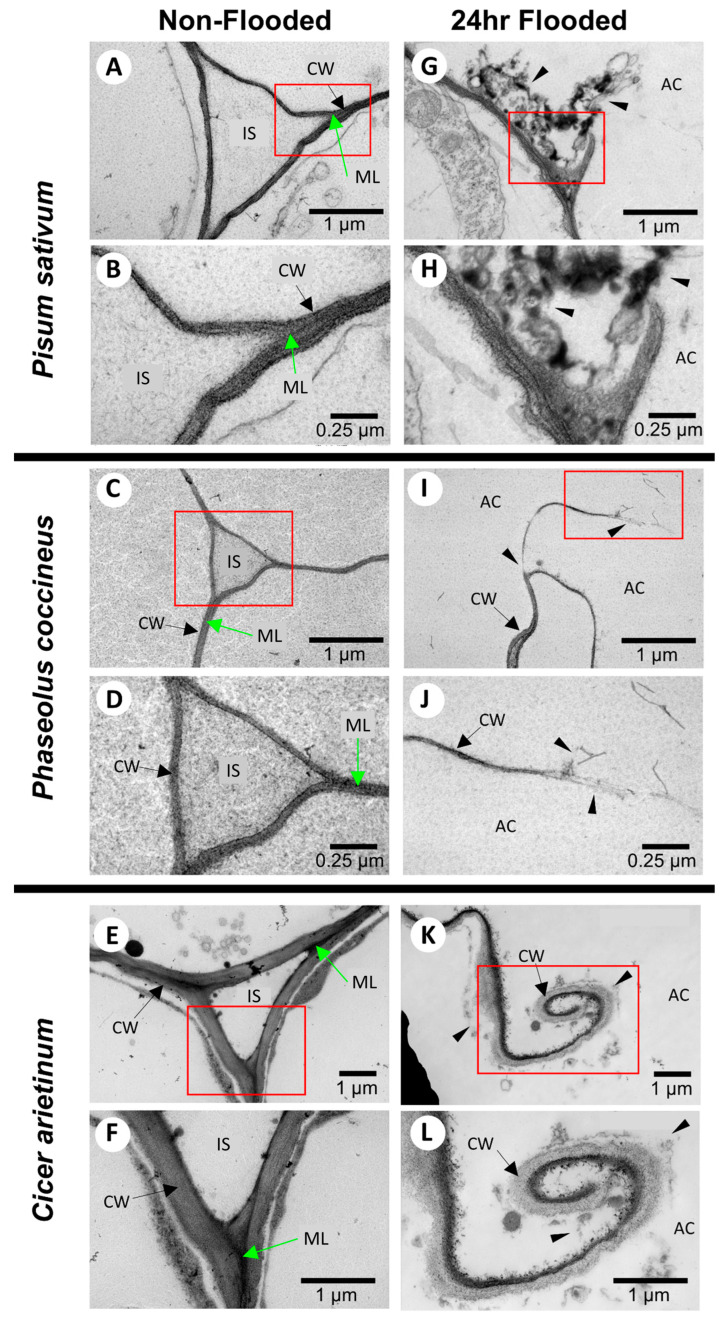
**TEM images of cell wall degradation during aerenchyma formation in legume roots.** Root cross-sections are stained with uranyl acetate, osmium tetroxide, and ruthenium red for (**A**–**F**) non-flooded samples at two magnifications for (**A**,**B**) *Pisum sativum*, (**C**,**D**) *Phaseolus coccineus*, and (**E**,**F**) *Cicer arietinum*. Opposite column displays comparison with (**G**–**K**) roots flooded for 24 h at two magnifications for (**G**,**H**) *P. sativum*, (**I**,**J**) *P. coccineus*, and (**K**,**L**) *C. arietinum*. IS = intracellular spaces, ML = middle lamella, AC = aerenchyma cavity, CW = cell wall. Red boxes show magnified areas. Black arrows indicate cell walls. Green arrows indicate middle lamella. Black wedges indicate areas of degraded cell wall components.

**Figure 2 plants-14-02620-f002:**
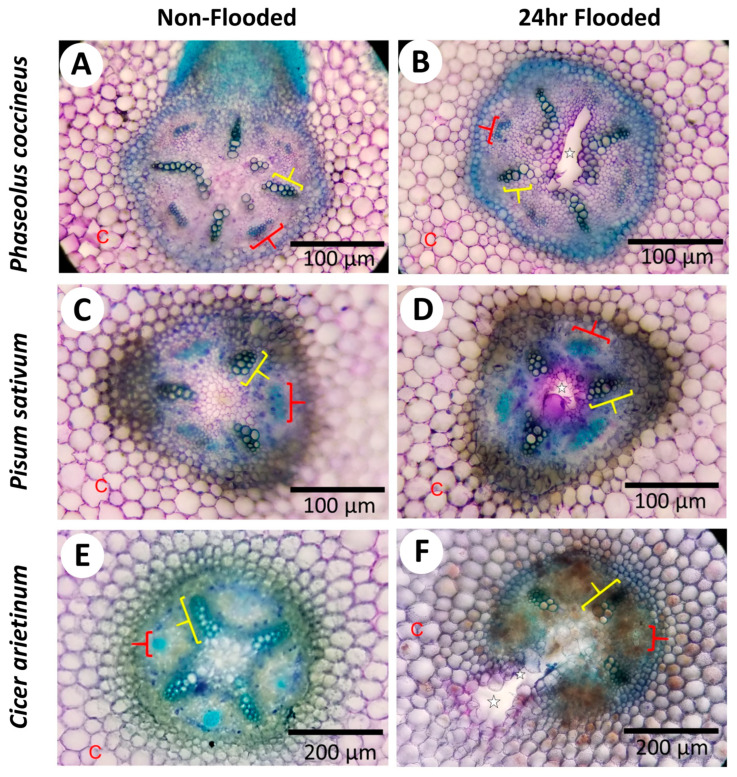
**Histochemical assays indicate alteration of pectin in cell walls of aerenchyma-forming legume roots.** Micrographs display cross-sections from *Phaseolus coccineus*, *Pisum sativum*, and *Cicer arietinum* roots exposed to (**A**,**C**,**E**) non-flooding and (**B**,**D**,**F**) 24 h flooding treatments. The aerenchyma cavities are indicated with white stars and wedges. Xylem cell walls and phloem sector fibers are indicated with yellow and red brackets, respectively. C = cortical cells.

**Figure 3 plants-14-02620-f003:**
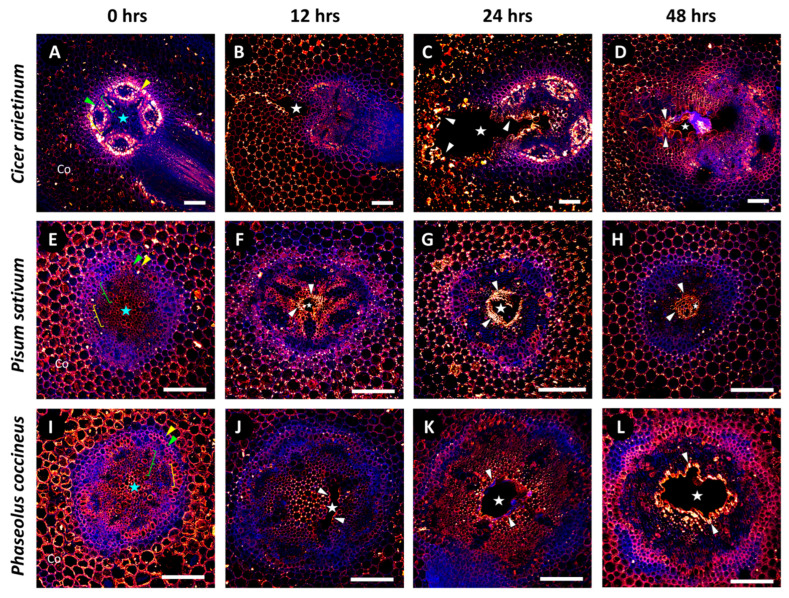
**Immunolabeling patterns for DMEH-calcium complexes (2F4) in legume root exposed to a 48 h flooding time course.** Micrographs display localization of 2F4 antibody labeling for (**A**–**D**) *Cicer arietinum*, (**E**–**H**) *Pisum sativum*, and (**I**–**L**) *Phaseolus coccineus*. The endodermis (yellow wedges), pericycle (green wedges), xylem cell walls (green brackets), phloem sector fiber (yellow brackets), and central parenchyma (blue stars) are indicated within the root cross-sections. Cell layers that are prominently labeled with antibodies are indicated with white wedges. Aerenchyma cavities have been indicated with white stars. Antibody labeling and aldehyde-enhanced fluorescence have been assigned red/yellow and blue false colors, respectively. Co = cortex. Scale bars = 100 µm.

**Figure 4 plants-14-02620-f004:**
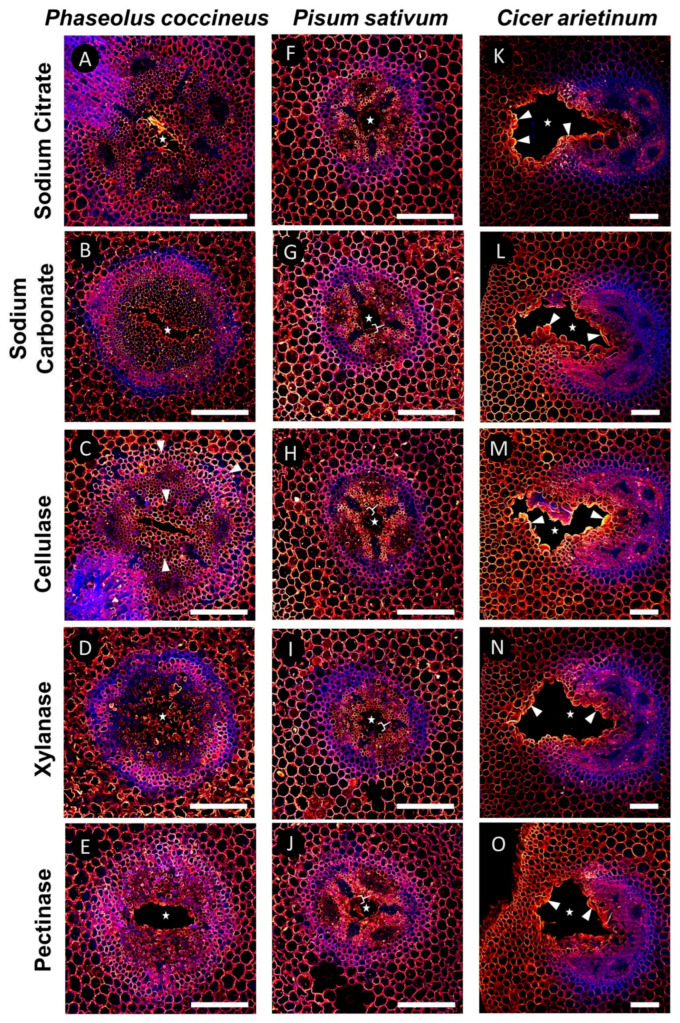
**Results of DMEH-calcium complex antibody labeling (2F4) in legume roots subjected to flooding stress.** (**A**–**E**) *P. coccineus*, (**F**–**J**) *P. sativum*, and (**K**–**O**) *C. arietinum* root cross-sections pretreated with (**A**,**F**,**K**) sodium citrate, (**B**,**G**,**L**) sodium carbonate, (**C**,**H**,**M**) xyloglucanase, (**D**,**I**,**N**) cellulase, and (**E**,**J**,**O**) pectinase enzyme solutions prior to labeling with 2F4 antibody. Antibody labeling and aldehyde-enhanced autofluorescence have been assigned red/yellow and blue false colors, respectively. The cell layers that are prominently labeled with antibodies are indicated with white wedges, while loss of labeling compared to sodium citrate treatment is indicated with white brackets. The white stars indicate aerenchyma cavities. Scale bars = 100 µm.

**Figure 5 plants-14-02620-f005:**
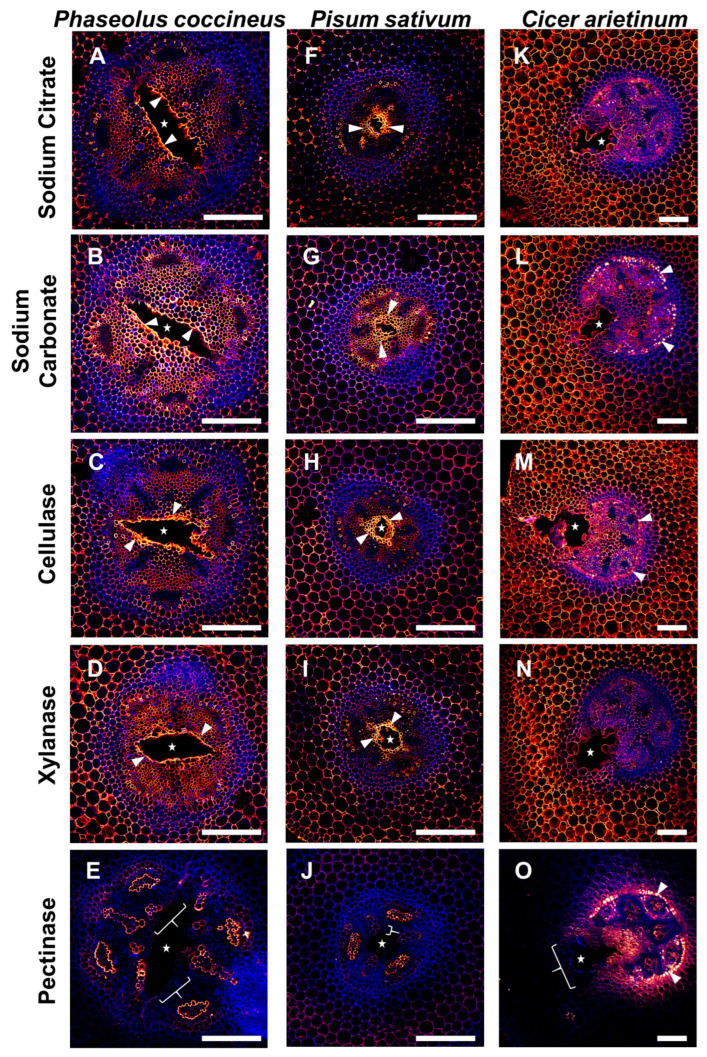
**Results of partially DMEH antibody labeling (JIM5) after enzymatic pretreatment of legume roots subjected to flooding stress.** (**A**–**E**) *P. coccineus*, (**F**–**J**) *P. sativum*, and (**K**–**O**) *C. arietinum* root cross-sections pretreated with (**A**,**F**,**K**) sodium citrate, (**B**,**G**,**L**) sodium carbonate, (**C**,**H**,**M**) cellulase, (**D**,**I**,**N**) xylanase, and (**E**,**J**,**O**) pectinase enzyme solutions. Antibody labeling and aldehyde-enhanced fluorescence were assigned red/yellow and blue false colors, respectively. The cell layers that are prominently labeled with antibodies are indicated with white wedges, while loss of labeling compared to sodium citrate treatment is indicated with white brackets. The white stars indicate aerenchyma cavities. Scale bars = 100 µm.

**Figure 6 plants-14-02620-f006:**
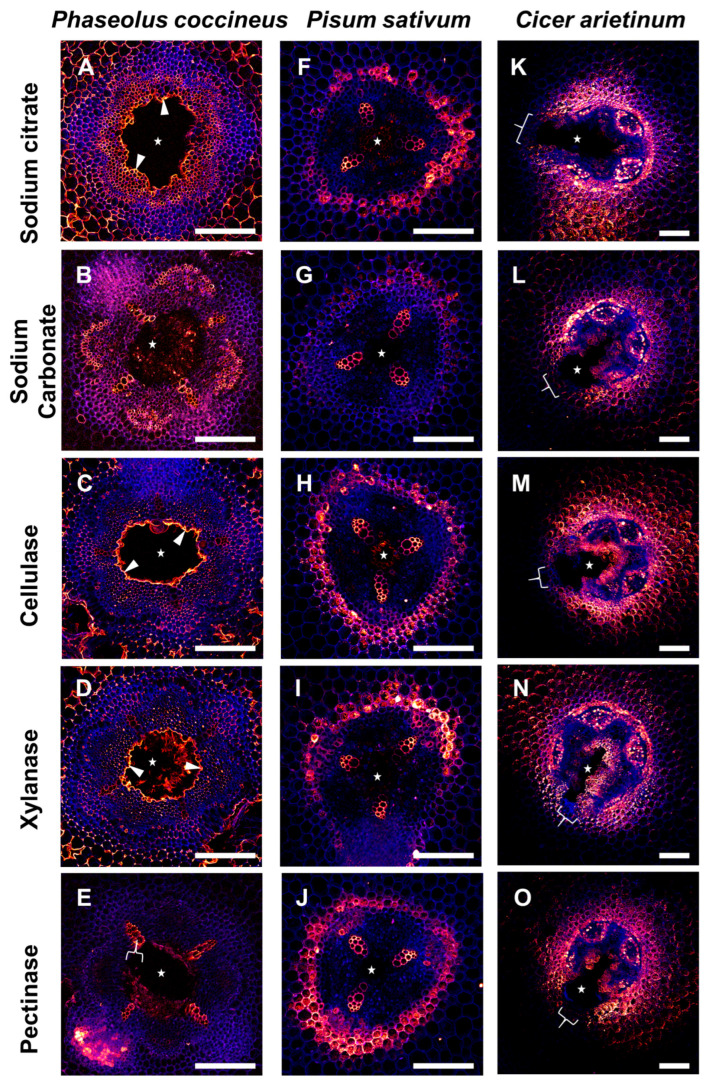
**Results of MEH antibody labeling (JIM7) in legume roots subjected to flooding stress.** (**A**–**E**) *P. coccineus*, (**F**–**J**) *P. sativum*, and (**K**–**O**) *C. arietinum* root cross-sections pretreated with (**A**,**F**,**K**) sodium citrate, (**B**,**G**,**L**) sodium carbonate, (**C**,**H**,**M**) cellulase, (**D**,**I**,**N**) xylanase, and (**E**,**J**,**O**) pectinase enzyme solutions. Antibody labeling and aldehyde-enhanced fluorescence were assigned red/yellow and blue false colors, respectively. The cell layers prominently labeled with antibodies are indicated with white wedges, while loss of labeling compared to sodium citrate treatment is indicated with white brackets. The white stars indicate aerenchyma cavities. Scale bars = 100 µm.

**Table 1 plants-14-02620-t001:** List of primary monoclonal antibodies.

Primary Antibody	Isotype	Source	Antigen
2F4	IgG1	Mouse	DMEH-calcium complex
JIM5	IgG2a	Rat	Partially MEH
JIM7	IgA	Rat	MEH

## Data Availability

The original contributions presented in this study are included in the article/Appendix A. Further inquiries can be directed to the corresponding author.

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
