# Peer review of "Pectin Peek-a-Boo: Homogalacturonan Turnover During Flooding-Induced Legume Root Aerenchyma Formation"

_plants, 2025, doi:10.3390/plants14172620_

Round 1
Reviewer 1 Report
Comments and Suggestions for Authors
The purpose of the research is interesting due to climate changes, cultivation of agronomical plants, and the use of immunohistochemical reactions to verify the changes occurring in plants.
However, I noticed methodological inaccuracies and missing controls that could support the stated hypothesis.
I would like the authors to highlight in the manuscript why the homogalacturonan (pectin) is the aim of the research, why this particular polysaccharide is so important in the studied process, why not other wall components such as hemicelluloses for example? Also I lack the highlighting the novelty or the importance of the undertaken study.
I have a lot of comments and questions, which I marked in attached pdf file. The comments that concern methods are of the most importance.

There are some not-scientific terms that should be replaced.
Author Response
Please see the attached document for a full list of comments.

Reviewer 2 Report
Comments and Suggestions for Authors
The present manuscript, “Pectin peek-a-boo: mechanisms for programmed cell death during flooding-induced legume root aerenchyma formation “investigated cell wall modifications during root aerenchyma formation in response to flooding in Phaseolus coccineus, Pisum sativum, and Cicer arietinum. The manuscript is written appreciably well; however, some changes are required before sending it for publication.
Lines 13-15 could be deleted from the abstract section. How were plants exposed to flooding? This information must be added in the abstract section.
Keywords are too long, revise the keywords and add only 5-6 reasonable keywords.
What could be the effects of flood stress on plants and root growth? The information about the underlying mechanisms from the latest studies must be supplemented in the updated version.
Lines 79-84: “Our results suggest that select cell walls bordering developing aerenchyma experience removal of cell wall components that leads to exposure and degradation of DME homogalacturonan required for aerenchyma formation. In addition, the presence of DME homogalacturonan throughout root stele tissues suggests de-methyl-esterification occurs independently of other cell wall unmasking events. This implies that de-methyl-esterification of homogalacturonan is required alongside other cell wall remodeling events for successful aerenchyma formation”. Please delete the following lines from the introduction section, as there is no need for these lines here.
Likewise, Lines 88-91 should also be deleted from the text, as there is no need of these lines. This is just explaining the methodological information. Therefore, I suggest the authors remove all this kind of information from the results section.
The results section needs to be revisited, and the authors should make this section more precise and concise. It is very extensive and contains much unnecessary information.
The discussion looks more review of literature than a discussion. Revisit this section and add cool and logical reasoning in this section.
Author Response
Comment 1: “What could be the effects of flood stress on plants and root growth? The information about the underlying mechanisms from the latest studies must be supplemented in the updated version.”
Response 1: Included a brief section concerning flooding effects and adaptations with regards to plants [Lines 36-47].
Comment 2: "Keywords are too long, revise the keywords and add only 5-6 reasonable keywords."
Response 2: Authors agree that some terms are required and necessary. Journal allows a limit of 10 keywords. We have reduced the terms to 7 words after discussion.
Comment 3: “Please delete the following lines [Lines 79-84] from the introduction section, as there is no need for these lines here. Likewise, Lines 88-91 should also be deleted from the text, as there is no need of these lines. This is just explaining the methodological information.”
Response 3: Authors agree with the reviewer’s comments and have deleted the relevant statements.
Comment 4: “The discussion looks more review of literature than a discussion. Revisit this section and add cool and logical reasoning in this section.
Response 4: Discussion section was significantly altered via deletions and revisions to phrasing. Remaining discussion section text was preserved at the collaborating authors’ request as it references peer-reviewed works that support conclusions drawn from this experiment In addition, the “review-like” aspect of the manuscript will be useful for the justification of continued research in future grant applications by the authors.
Round 2
Reviewer 1 Report
Comments and Suggestions for Authors
Thank You for taking my remarks into consideration. Still have some.
I wanted to kindly follow up on the question I sent (how did You verify whether enzymatic removal of these components was effective? were components actually removed? these data should be attached in supplementary; to show that prior to immunolabelling the enzymatic treatment worked). You did not provide answer for it in the cover letter. I was hoping for a response, and I must admit I’m a bit disappointed not to have received one. I know that You plan more publications so please focus on controls in the future. You have to prove that something You remove, it is not there anymore.
And actually Your response to comment 8 is better than what it is put in the manuscript. Good aim for study.

Just check the remarks. Some descriptions are not scientific. And the use of abrreviations is not consequent. Figures description, stars; are also not consistent.
Author Response
Dear Reviewer,
Thank you for your continued evaluation of our manuscript. Your time and commentary is greatly appreciated by the authors. Below is a detailed list of corrections made to the manuscript (line #'s based on previous PDF file sent by the Reviewer).
Comment: Follow-up; How did You verify whether enzymatic removal of these components was effective? Were components actually removed?
Authors' Response: In the current experiment, the enzyme removal was implied by the changes in antibody labeling patterns between treatments as compared to the negative control (for enzyme activity) - sodium citrate (standard buffer solution for each enzyme) - and the positive control (for antibody binding to DMEH) - sodium carbonate (removal of methyl ester groups from exposed homogalacturonan). In addition, the treatment of pectinase (Example: Figure 5) greatly diminished the effectiveness of JIM5 binding (partially DMEH), demonstrating this enzyme solution was activity changing epitope availability - likely through thwe removal of homogalacturonan residues.
However, the reviewer makes an excellent point that will be addressed in follow-up research design. A suggestion has been to utilize a 96-well plate format and expose plant tissue samples to each enzyme, followed by testing with kits measuring cell wall breakdown products. For example, a modification of the Megazyme Cellulose Assay kit protocol (https://www.megazyme.com/Cellulose-Assay-Kit-CELLG5-Method) could be utilized to evaluate cellulase enzyme activity on our samples. As the reviewer suggests, this would be conducted for each cell wall component evaluated (cellulose, xylan, pectins, etc.) and included in a discussion section.
While commercially available during the time of this study, research funding was not available for multiple test kits during the primary authors' Ph.D. studies that generated this data set. Additionally, COVID-19 pandemic restrictions greatly impacted equipment availability (e.g. plate reader) required for enzyme kit evaluation.
List of changes:
- Line 22 - Used the "MEH" abbreviation for methyl-esterified homogalacturonan
- Line 52 - Corrected text to "Arenchymas"
- Line 71 - Used "DME" abbreviation for de-methyl-esterification
- Line 90 - Corrected to, "Our hypothesis was that …"
- Line 92 - Used the DME abbreviation
- Line 93-94 - Clarified stated at the reviewer's request
- Line 96, 98 - Adopted reviewer suggestions
- Line 101 - Used the "DMEH" for de-methyl-esterified homogalacturonan
- Figure 1 - Add arrows from ML to the middle lamella
- Corrected the location of middle lamella notation (ML) and added "IS" notation to represent intracellular spaces.
- Used green arrows to indicated ML location inside micrographs
- Line 119, 127 - Adopted review corrections
- Figure 2 - Changed to "24hr flooding and Non-flooded"
- Reversed positions of images and altered lettering in the final manuscript
- Line 133 - remove the "_" beside (A, C, E)
- Figure 3 - Corrected figure image to show proper location of endodermis and pericycle
- Added blue stars to show the center of root sections at "0 hrs"
- Enlarged white stars 100% whenever possible to indicate aerenchyma cavities - some aerenchyma cavities would be covered by enlarged stars
- Line 139 - Removed the sentence highlighted by the reviewer to improve clarity and conciseness of the subsequent paragraph.
- Line 151-154 - Changed results sentence, per reviewer #1 recommendation
- Line 175 - Adopted reviewer corrections regarding "domain" usage in the sentence.
- Line 177 - Specified that the authors were evaluated antibody labelling of cell walls both nearby and further away from the aerenchyma cavity.
- Line 192-193 - Change abbreviation to DMEH
- Line 194-195 - Adopted the recommendation to move this sentence to the the discussion section.
- Line 210 - Used "artificially-induced" instead of "forced" in this sentence.
- Line 216 - Used "Did not increase" in this sentence.
- Figure 5 - Rewrote the legend in a similar manner to Figure 4
- Line 239 - Adopt reviewer recommendation: abbreviation of MEH
- Line 251 - Adopt reviewer recommendation: Changed ME homogalaturonan to MEH
- Lines 256 - Adopt reviewer recommendation: Usage of MEH without writing it out.
- Figure 6 - Rewrite the legend in a similar manner to Figure 4
- Line 266 - Used the DMEH abbreviation
- Line 270 - Adopt reviewer recommendation: Insertion of "and both cellulase and xylanase pretreatments resulted in increased JIM7 labeling (...)" into the sentence
- Line 283 - He wants me to specified cleavage by pectic enzymes such as pectin lyase and polygalacturonase
- New reference #35 included to support modifications to the sentence.
- Sénéchal F.; Wattier C.; Rustérucci C.; Pelloux J. Homogalacturonan-modifying enzymes: structure, expression, and roles in plants. J Exp Bot. 2014, 65, 5125-5160.
- Lines 285, 287 292 - Adopted reviewer recommendation: grammatical and spelling changes.
- Line 299, 301, 309, 324 - Used the DMEH abbreviation
- Line 325 - Used the word "completely" instead of "fully" - more scientific.
- Line 331 - Used the phrase, "may be masked" in this sentence
- Line 359 - Adopt the review comment and used it to modify this sentence.
- "... You found evidence that selective removal of cellulose and hemicelluloes may have implication on pectin detection; and this procedure may mimic the processes that occur during cavity formation."
- Line 359 (2) - Used the phrase "selected" instead of "select"
- Line 416 - Used the word, "with" - it was previously missing in this sentence.
- Line 418 - Used the word, "utilized".
Reviewer 2 Report
Comments and Suggestions for Authors
The manuscript can be accepted for publication.
Author Response
Dear reviewer,
Thank you for your time and consideration of our manuscript. Please let us know if you have any additional questions or concerns.